# The Brief Measure of Emotional Preoperative Stress (B-MEPS) as a new predictive tool for postoperative pain: A prospective observational cohort study

Anelise Schifino Wolmeister[1,2], Carolina Lourenzon Schiavo[1], Kahio César Kuntz Nazário[3], Stela Maris de Jezus Castro[4], Andressa de Souza[2], Rafael Poli Caetani[5], Wolnei Caumo[6], Luciana Cadore Stefani[2,3,6]*

1 Postgraduate Program in Medical Sciences, School of Medicine, Universidade Federal do Rio Grande do Sul (UFRGS), Porto Alegre, Brazil, 2 Laboratory of Pain & Neuromodulation, Hospital de Clínicas de Porto Alegre (HCPA), Porto Alegre, Brazil, 3 Anaesthesia and Perioperative Medicine Service, Hospital de Clínicas de Porto Alegre (HCPA), Porto Alegre, Brazil, 4 Department of Statistics, Universidade Federal do Rio Grande do Sul (UFRGS), Porto Alegre, Brazil, 5 School of Medicine, Universidade Federal do Rio Grande do Sul (UFRGS), Porto Alegre, Brazil, 6 Department of Surgery, School of Medicine, Universidade Federal do Rio Grande do Sul (UFRGS), Porto Alegre, Brazil

* lpstefani@hcpa.edu.br

**Data Availability Statement:** All relevant data are within the manuscript and its support information files.

## Abstract

### Background

Preoperative patients' vulnerabilities such as physical, social, and psychological are implicated in postoperative pain variability. Nevertheless, it is a challenge to analyze a patient's psychological profile in the preoperative period in a practical and consistent way. Thus, we sought to identify if high preoperative emotional stress, evaluated by the Brief Measure of Emotional Preoperative Stress (B-MEPS) scale is associated with higher postoperative pain levels and poor rehabilitation in patients submitted to intermediate or major surgery. Moreover, the possible neurobiological or neurophysiological mechanisms implicated in high preoperative emotional stress, evaluated through preoperative quantitative sensory pain tests and serum biomarkers BDNF and S100B were investigated.

### Methods

We conducted a prospective, observational, cohort study of ASA 2 and 3 adult patients undergoing major urologic, gynecologic, proctologic and orthopedic surgeries from March 2017 to March 2018. B-MEPS and Central Sensitivity Inventory were evaluated preoperatively, followed by a sequence of experimental pain tests and serum biomarkers collection. Postoperative evaluation carried out within the first 48 hours after surgery comprehended pain at rest and movement-evoked pain, and the consumption of morphine. Quality-of-Recovery was also evaluated in the 3rd postoperative day.

### Results

23 (15%) out of 150 patients included in the study presented high emotional preoperative stress. Variables significantly related to preoperative stress were: previous psychiatric

**Funding:** Research reported in this publication was supported by Research and Events Incentive Fund - Hospital de Clínicas de Porto Alegre (FIPE-HCPA), Porto Alegre, Brazil (Application 170091); Postgraduate Program in Medical Sciences, School of Medicine, Universidade Federal do Rio Grande do Sul, Porto Alegre, Brazil; Postgraduate Research Group, HCPA, Porto Alegre, Brazil. The funding sources had no influence in the study design, data collection and analyses, manuscript preparation, or in the decision to submit the article for publication.

**Competing interests:** The authors have declared that no competing interests exist.

diagnosis and Central Sensitization Inventory result. Mean movement-evoked pain in the first 12 to 48 hours was 95–105% higher than pain at rest. A mixed model for repeated measures showed a sustainable effect of B-MEPS as a movement-evoked pain predictor. Previous pain, cancer surgery, and preoperative pressure pain tolerance were also independent predictors of postoperative pain. Moderate to severe postoperative movement-evoked pain was predictive of poor rehabilitation in 48 hours after surgery.

## Conclusion

We confirmed that a brief screening method of preoperative emotional states could detect individuals prone to experience severe postoperative pain. Specific interventions considering the stress level may be planned in the future to improve perioperative outcomes.

## Introduction

Postoperative moderate to severe pain remains one of the major concerns in spite of considerable advances in pain prevention, treatment, and management. It is worrisome that a considerable number of patients still experience moderate to severe postoperative pain in major [1] or even minor surgeries [2] since it has an impact in many important outcomes such as increased rates of organ dysfunctions [3], longer hospital stays and costs. Also, acute postoperative pain is associated with poor satisfaction and rehabilitation [4] and increases the risk of developing postoperative chronic pain [5]. A huge variation in postoperative pain thresholds is frequently observed in the postoperative scenario, even for similar surgical trauma and type of anesthesia. Preoperative patients' vulnerabilities such as physical, social, and psychological are implicated in this variability [5]. Nevertheless, analyzing, in a practical and consistent way, a patient's psychological profile in the preoperative period constitutes a challenging task, even considering the existing evidence of worst rehabilitation outcomes related to catastrophizing levels [6], anxiety [7], surgical fear [8] and preoperative pain [8,9]. In general, a good psychological state is a health indicator. Healthy psychological aspects such as life satisfaction, optimism, self-esteem, and perception of social support can positively influence several health indices. On the other hand, factors such as depression, anxiety, hostility reflect a less desirable psychological state, which may influence short and long-term recovery in direct and indirect ways. The direct effect is related to the impact of emotions on the stress hormones (cortisol, adrenaline, noradrenaline) which regulate healing and many physiological responses. Besides, the indirect effect of psychological burden can be reflected on behavioral responses to stress such as poor self-care, smoking, alcohol intake, anxiety, depression and sleep deprivation [10–12].

Our rationale is that surgery can be considered a powerful external stressor, a major life event, causing in the organism a cascade of physiological and psychological reactions as protective, coordinated and adaptive responses [13,14]. Stress occurs when environmental stressors go beyond acceptable levels and disturbs the adaptive capacity (allostasis) [15]. We have recently developed an instrument to assess individual psychological vulnerability based on emotional stress in the preoperative period, the Brief Measure of Emotional Preoperative Stress (B-MEPS) [16].

In order to add a feasible instrument to carry out a brief and thorough preoperative psychological evaluation, the item response theory strategy was successfully used to group significant items from four different tools currently used to measure depression and anxiety symptoms,

minor psychiatric problems and future self-perceptions [16] such as State-Trait Anxiety Inventory (STAI), the Montgomery-Asberg Depression Rating Scale (MADRS), the World Health Organization's (WHO) Self-reporting Questionnaire (SRQ-20), and the Future Self-perception Questionnaire (FSPQ). Therefore, more than being a sole psychological scale designed to measure just one symptom (depression or anxiety), the B-MEPS instrument aims at adding a broad emotional evaluation to the preoperative setting. However, the neurophysiological and biological mechanisms possibly related to the emotional preoperative stress and its consequences need to be explored for a better understanding of the preoperative patient's profile as a whole.

With a view to facing this challenge, this study has three aims: (1) investigate the possible neurobiological or neurophysiological mechanisms implicated in high preoperative emotional stress, evaluated through preoperative quantitative sensory pain tests and serum biomarkers BDNF and S100B (2) identify the relation between preoperative stress and the postoperative movement-evoked pain and morphine consumption during the first 48 hours (3) analyze the influence of preoperative stress on postoperative rehabilitation. We hypothesized that high preoperative stress, evaluated by B-MEPS measure, is associated with elevated experimental preoperative pain thresholds and also higher postoperative pain in patients undergoing intermediate or major surgery.

## Methods

### Setting and patients

The project was submitted to and approved by the research ethics committee of Hospital de Clínicas de Porto Alegre (Application 170091), and written consent was obtained for all participants. We conducted a prospective, observational cohort study of ASA 2 and 3 in adult patients undergoing major urologic, gynecologic, proctologic, and orthopedic surgeries from March 2017 to March 2018 at Hospital de Clínicas de Porto Alegre, Rio Grande do Sul. One hundred fifty-three patients were sequentially recruited, and written consent was obtained.

Exclusionary criteria were medical history of brain damage, history of intellectual disability or cooperation incapacity and personal refuse.

### Study overview

All preoperative assessments were conducted by the main investigators (ASW, CLS). A schematic illustration of the study protocol is shown in Fig 1.

### Preoperative

Eligible patients were approached and provided written informed consent the night before surgery. Trained researchers collected the demographic data, basal health questionnaire, the B-MEPS scale (Table 1), and Central Sensitization Inventory [17]. Data on the clinical and psychiatric comorbidities, the reason for surgery, the presence of neoplasia, and the intensity of acute or chronic pain were also evaluated. The use of psychotropic and pain control medications was registered.

B-MEPS assessment: The B-MEPS tool was refined and cut-off points were identified for categorizing patients according to the intensity of preoperative psychological stress. A reduced version with 12 items was applied to classify patients as undergoing low stress (up to 0.22 SD above average), intermediate stress (between 0.22–0.77 SD), or high stress (above 0.77 SD). An interface for practical use and bedside application was developed using Shiny applications, which is a package from *RStudio*. The tool is available at https://rogerio.shinyapps.io/r_shiny/

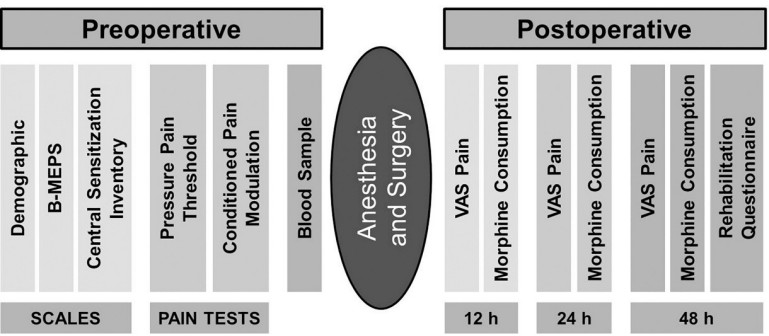

**Fig 1. Schematic illustration of the study protocol.** The day before scheduled surgery patients completed the Demographic, B-MEPS tool, and Central Sensitization Inventory (CSI). Then, patients underwent a series of experimental pain tests: Pressure pain threshold (PPT), pressure pain tolerance (PPTol) and conditioned pain modulation test (CPM) and blood sample analysis. During postoperative days 1–3 pain-related outcomes were assessed, including postoperative rest pain and movement-evoked pain intensity, and morphine Rehabilitation Questionnaire (QoR-15) was applied on the 3rd day.

and could be accessed for research and clinical practice purposes. The questions are illustrated in Table 1.

**Preoperative experimental pain tests.** Pressure pain threshold (PPT)—After a brief explanation, an experimental pain test was performed using a digital pressure algometer device with a 1cm$^2$ probe (Biolink). Pressure algometry was performed on the patient's dominant forearm, 3 cm from the cubital fossa. The pressure was gradually increased at a rate of 0.3 Kgf per second. We asked patients to differentiate the perception of pressure from the perception of the onset of pain. At this moment, the pressure measured in Kgf on the algometer display was registered. This represents the pressure pain threshold. Three successive readings taken at intervals of 3 to 5 minutes were used to define the PPT in kgf/cm$^2$ (lb/cm$^2$). After that, the pressure stimuli were gradually increased to the maximum tolerable level and the pain pressure tolerance was then registered.

**Table 1. The refined version of B-MEPS tool.** Instruction to patients: "These questions aim to assess your feelings of stress related to the perioperative period".

| | Item content | Response scale | | | |
|---|---|---|---|---|---|
| **1.** | I am jittery | (1) not at all | (2) somewhat | (3) moderately | (4) very much so |
| **2.** | I feel indecisive | (1) not at all | (2) somewhat | (3) moderately | (4) very much so |
| **3.** | I am worried | (1) not at all | (2) somewhat | (3) moderately | (4) very much so |
| **4.** | I feel confused | (1) not at all | (2) somewhat | (3) moderately or very much so | |
| **5.** | I feel like a failure | (1) almost never | (2) often | (3) almost always | |
| **6.** | I worry too much over something that really doesn't matter | (1) almost never | (2) often | (3) almost always | |
| **7.** | I take disappointments so personally that I can't get them out of my mind | (1) almost never | (2) often | (3) almost always | |
| **8.** | I get in a state of tension or turmoil as I think over my recent concerns and interests | (1) almost never | (2) often | (3) almost always | |
| **9.** | Do you feel unhappy? | (1) No | (2) Yes | | |
| **10.** | Do you have feelings of discomfort in the stomach? | (1) No | (2) Yes | | |
| **11.** | How do you react when you are unhappy? | (1) I may look dispirited but I brighten up easily | | | |
| | | (2) I have pervasive feelings of sadness or feel constantly gloomy | | | |
| **12.** | How do you describe your depressed mood? | (1) Ocasional sadness | | | |
| | | (2) External factors can change it | | | |
| | | (3) Being without help or hope | | | |

B-MEPS, Brief Measure of Emotional Preoperative Stress.

Following, patients were instructed to identify when the pressure represented a pain between 6 and 10 in a visual analog scale (VAS) and those values were registered.

**Conditioned pain modulation test.** The integrity of the endogenous inhibitory system was evaluated using Conditioned Pain Modulation (CPM) test. To apply the CPM, we used the Tousignant-Laflamme et. al. protocol [18]. The experimental pain stimulus used was in accordance with the guidelines for the cold-pressor task (CPM-task). The CPM-task allows us to modify the descendent pain modulatory system. In order to perform it, a heterotopic stimulus was induced and the pain threshold difference between before and after the application of the stimulus was measured. For this, we used the technique of immersing the patient's hand in cold water with controlled temperature for a maximum of 30 seconds or until the patient no longer bears the pain. At the same time, a pressure algometry equivalent to 6 in VAS was performed, based on the patient's previous report. The patient is then asked to verbally report the pain intensity and discomfort associated with the immersion by evaluating it numerically, based on the visual analog pain scale (0–10) and algometry pressure forearm. The difference between the new value in dominant arm and 6 was then calculated. Whenever the result was negative, there was pain facilitation (modulator deficient system), while positive scores represented system integrity (inhibition of pain by the conditioning stimulus).

**Blood sample analysis.** Blood sample was collected and later processed and stored to analyze serum S100B protein and brain-derived neurotrophic factor (BDNF). The blood was centrifuged for 10 minutes at 4500 rpm at 4°C. Serum was frozen at -80°C until further analysis. The concentration of serum mediators S100B (Millipore, Missouri, USA, catalogue #EZHS100B-33 K, kit's lower detection limit = 2.7 pg/mL) and BDNF (Millipore, Missouri, USA, catalogue #CYT306, kit's lower detection limit = 7.8 pg/mL) was then determined using enzyme-linked immunosorbent assay (ELISA) kits, according to the manufacturer's instructions.

## Transoperative

There was no interference in the decision of the anesthetic technique adopted. Patients were submitted to general or combined anesthesia either with spinal or epidural anesthesia. The type and amount of opioid administered intraoperatively were recorded.

## Postoperative

Postoperative analgesia was carried out with multimodal analgesia. When epidural analgesia was provided, the infusion consisted of 0.125% bupivacaine for the first 48 hours. Morphine was prescribed to all patients suffering of uncontrolled pain, besides non-opioid analgesic treatment with acetaminophen (4g a day) and dipyrone (4g a day), or anti-inflammatory- if not contraindicated.

**Pain assessment.** Pain at rest or movement-evoked pain was assessed by a visual analog scale from zero (absence of pain) to 10 (worst possible pain) in 12, 24 and 48 hours in the postoperative period. Also, the morphine consumption was evaluated during the first 48 hours.

**Rehabilitation.** The Quality of Rehabilitation Questionnaire (QoR-15) was applied within the 48 hours of the postoperative period. The score ranges from 0 to 150, with a higher score indicating a better rehabilitation. It measures the domains of pain, physical comfort, physical independence, psychological support, and emotional state. The original QoR-15 has been translated into Portuguese and validated [19]. All data were collected using the shared electronic platform Redcap. The sequence of study protocol is resumed in Fig 1.

**Statistical analysis.** Demographic data are presented as the mean ± standard deviation (SD), median (interquartile range), number (%) or 95% confidence interval (CI). Preoperative

stress was calculated for everyone using the B-MEPS new version. Crude exploratory associations between variables were measured using Spearman rank (ρ) correlation coefficients. To explore preoperative variables possibly associated to preoperative stress as continuous variables, a Generalized Linear Model was run with B-MEPS results as dependent variables and demographic variables related to basal psychological status and biomarkers as independent variables.

To test the null hypothesis that high preoperative emotional stress influences postoperative pain, a MANCOVA analysis was run with preoperative pain test results and postoperative pain measures such as pain at rest and movement-evoked pain as dependent variables. The model was adjusted to covariates, which could interfere somehow in the postoperative pain measurement (age, schooling, and cancer surgery).

The impact of preoperative stress in postoperative pain was evaluated considering the cut-off points of B-MEPS. A mixed model for the repeated measurements was done to verify differences in movement-evoked pain using time as a factor. We included variables according to biological plausibility stepwise until the final model was achieved.

For the analysis of secondary outcomes, such as predictors of morphine consumption within 48 hours and the Rehabilitation Result (QoR-15 result), we performed two Generalized Linear Models.

Sample size was estimated with G*Power 3.1 in 120 patients considering a medium effect size (F = 0.15) of high stress in postoperative pain using a MANOVA model for repeated measures, with a power of 0.8 and alfa 0.05. For all statistical analyses, the significance was set at $p < 0.05$. For the B-MEPS result calculation, the R statistic program version 3.2.3 was used and for correlation, a regression analysis on the SPPS version 22.0 was used.

## Results

Fig 2 shows the flow chart of the study in which 150 patients from the urology, proctology, traumatology and gynecology units from Hospital de Clínicas de Porto Alegre were included and followed after surgery. Complete descriptive statistics for demographic and clinical characteristics, preoperative predictors, and postoperative outcomes are depicted in Table 2.

### Crude association of preoperative and transoperative predictors for pain and stress

Regarding potential predictors or confounding variables, Spearman's ranking of correlations for preoperative, transoperative and postoperative predictors for the following outcomes are shown in Table 3: B-MEPS result, acute postoperative pain or rehabilitation. Notably, there was a moderate correlation between the B-MEPS result and the Central Sensitivity Inventory (CSI) (r = 0.506; p < 0.01). The biomarker BDNF was inversely correlated with morphine consumption and length of stay. Morphine consumption in 48 hours was positively associated with the duration of surgery (r = 0.25; p = 0.03). We found no evidence of the association between QoR-15 rehabilitation questionnaire result in 48 hours and preoperative stress.

### Predictors of preoperative emotional stress

A generalized linear model was run to analyze the preoperative predictors of emotional preoperative stress (B-MEPS) result. The variables included in the model explained 28% of the B-MEPS values. Preoperative psychiatric diagnosis ($p = 0.012$) and CSI results ($p < 0.01$) were the only significant preoperative predictors in the model (Table 4).

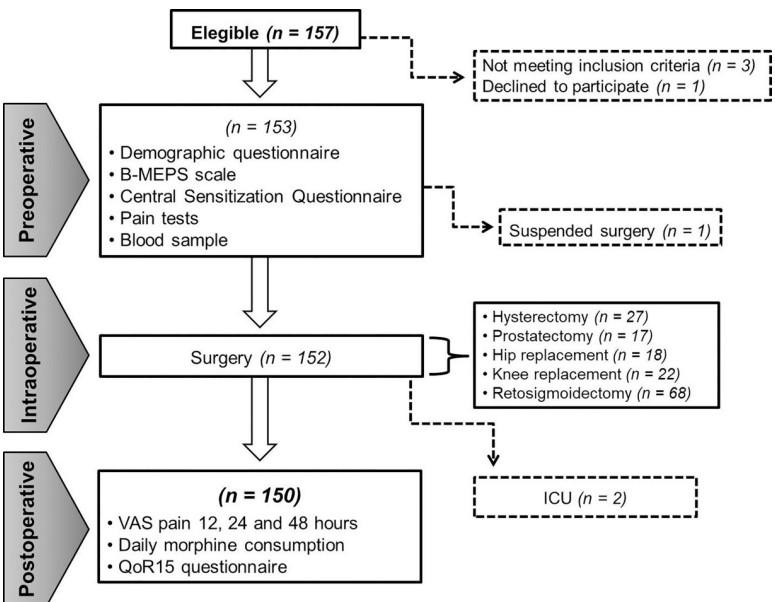

**Fig 2. Flow chart of the study.**

## Primary outcomes

Patients were categorized according to the intensity of stress: low stress [up to 0.22 SD above average], intermediate stress (between 0.22–0.77 SD) or high stress (above 0.77 SD). With latent class analysis we could identify 23 patients (15%) with high stress.

Preoperative pain tests and postoperative pain measures were compared between the groups of high or low-stress patients with MANCOVA, controlling for age, schooling, and cancer surgery (Table 5). The dependent variables included in the analysis had a normal distribution and the Levene test for variance homogeneity was not significant. We found a higher numerical pain scale in movement-evoked pain in 12 and 24 hours in high-stressed patients. No difference between preoperative pressure pain threshold, pressure pain tolerance, or conditioned pain modulation was found.

To thoroughly explore postoperative movement-evoked pain predictors, we ran a mixed model for repeated measures depicted in Table 6. B-MEPS was added to the model as a fixed effect, and time as a repeated measure. We found a moderate effect size of B-MEPs of 0.46 (standard mean difference) on movement-evoked pain. Preoperative pain, cancer diagnosis, and pressure pain tolerance were also significant variables in the final model.

Also, the predictor factors of morphine consumption in 48 hours in the postoperative period was analyzed (Table 7) in a variance analysis after residual confirmation. High stress, general anesthesia and cancer were predictors to higher postoperative morphine consumption. The morphine consumption means in the high-stress patients was 11.46 (SD 1.4) and 8.10 mg (SD 0.9) for intermediate and low-stress group.

## Postoperative rehabilitation

Finally, the predictors of rehabilitation were evaluated using the QoR-15 scale, which provides a global, patient-centered measure of recovery after anesthesia and surgery as a dependent variable. Independent variables included in the analysis were B-MEPS, schooling, cancer surgery, type of anesthesia, postoperative pain, and surgical duration. Postoperative movement-evoked pain at 48 hours was the only significant predictive variable (F = 15.69; $p < 0.001$) depicted in Table 8.

**Table 2. Demographic and clinical characteristics, preoperative predictors and postoperative outcomes (n = 150).**

| Variables | *n* = 150 |
|---|---|
| Age (mean ± SD) | 58.2 ± 12.1 |
| Gender (female) | 87 (58%) |
| Smoking | 16 (10.7%) |
| Alcohol intake | 18 (12%) |
| Chronic pain | 33 (22%) |
| Chronic pain medication | 7 (4.7%) |
| Psychiatric diagnosis | 31 (20.7%) |
| Cancer diagnosis | 77 (51.3%) |
| Schooling (P25-P75) | 8 (5–11) |
| **Preoperative pain test** | **Mean ± SD** |
| Pressure pain threshold | 2.83 (1.46) |
| Pressure pain tolerance | 7.48 (2.94) |
| Reduction of pain scale on conditioned pain modulation task | -1.46 (2.94) |
| **Preoperative questionnaires** | **Mean ± SD** |
| CSI | 29.41 (15.71) |
| B-MEPS | -0.21 (-0.3) |
| **Anesthesia and surgery** | **Mean ± SD** |
| General anesthesia | 16 (10.7%) |
| Morphine neuraxial | 121 (80.7%) |
| **Procedures** | **Mean ± SD** |
| Hysterectomy | 27 (18%) |
| Prostatectomy | 17 (11.3%) |
| Hip replacement | 16 (10.7%) |
| Knee replacement | 22 (14.7%) |
| Retosigmoidectomy | 68 (45.3%) |
| **Pain related outcomes** | **Mean ± SD** |
| Pain at rest 12 h | 2.88 (2.98) |
| Pain at rest 24 h | 5.64 (3.07) |
| Pain at rest 48 h | 2.78 (2.76) |
| Movement-evoked pain 12 h | 3.51 (3.16) |
| Movement-evoked pain 24 h | 6.33 (2.83) |
| Movement-evoked pain 48 h | 5.71 (2.82) |
| **Morphine consumption** | **mg** |
| Morphine consumption 12 h (P25—P75) | 0–4 |
| Morphine consumption 24 h (P25—P75) | 0–4.75 |
| Morphine consumption 48 h (P25—P75) | 0–3 |
| Rehabilitation (QoR15) | 108.06 (21.55) |

SD, Standard Deviation; B-MEPS, Brief Measure of Emotional Preoperative Stress; CSI, Central Sensitivity Inventory; QoR-15, Quality of Rehabilitation Questionnaire-15.

## Discussion

This study assessed whether preoperative emotional stress evaluated with a new tool—the B-MEPS—can predict postoperative outcomes. It was implemented in a cohort of patients undergoing major proctologic, orthopedic, gynecologic or urologic surgeries. We can highlight three related main findings. First, we reinforced the framework of perioperative stress

**Table 3. Spearman's rank correlation matrix for preoperative, transoperative, and postoperative variables.**

| Variable | 1. B-MEPS value | 2. CSI | 3. Pressure pain threshold | 4. Pressure pain tolerance | 5. CPM | 6. BDNF | 7. S100B | 8. Surgical length | 9. NPS (0–10) Movement pain 48 h | 10. Morphine consumption 48 h (mg) | 11. QoR-15 | 12. Length of stay |
|---|---|---|---|---|---|---|---|---|---|---|---|---|
| 1. B-MEPS value | 1 | | | | | | | | | | | |
| 2. CSI | **0.506** | 1 | | | | | | | | | | |
| 3. Pressure pain threshold | -0.03 | **-0.289** | 1 | | | | | | | | | |
| 4. Pressure pain tolerance | -0.06 | **-0.202** | **0.488** | 1 | | | | | | | | |
| 5. CPM | 0.026 | 0.013 | -0.151 | -0.105 | 1 | | | | | | | |
| 6. BDNF | -0.02 | -0.022 | 0.093 | 0.094 | 0.118 | 1 | | | | | | |
| 7. S100B | -0.07 | 0.107 | 0.017 | -0.054 | -0.032 | -0.039 | 1 | | | | | |
| 8. Surgical length | -0.03 | -0.04 | 0.089 | 0.07 | -0.109 | **-0.193** | -0.129 | 1 | | | | |
| 9. NPS (0–10) Movement pain 48 h | 0.06 | **0.208** | 0.068 | -0.045 | -0.038 | -0.106 | -0.139 | 0.137 | 1 | | | |
| 10. Morphine consumption 48h (mg) | 0.01 | 0.004 | 0.026 | 0.086 | 0.046 | **-0.193** | -0.089 | **0.25** | **0.185** | 1 | | |
| 11. QoR-15 | -0.08 | -0.155 | 0.057 | 0.064 | 0.011 | 0.0 | 0.114 | -0.013 | -0.15 | -0.085 | 1 | |
| 12. Length of stay | -0.06 | -0.86 | -0.137 | 0.005 | -0.003 | **-0.195** | -0.044 | **0.407** | -0.012 | **0.268** | 0.00 | 1 |

Spearman's rank correlation coefficients for all pairs of preoperative, perioperative, and postoperative study variables including predictors, confounders and outcomes. Total *n* for the pairwise correlations with pre- and intraoperative variables is 150. Coefficients with significance levels of 0.05 or less are printed in bold. B-MEPS, Brief Measure of Emotional Preoperative Stress; CSI, Central Sensitivity Inventory; CPM, Conditioned Pain Modulation; BDNF, brain-derived neurotrophic factor; NPS, Numeric Pain Scale; QoR-15, Quality of Rehabilitation Questionnaire– 15.

measured by the B-MEPS tool. Second, our hypothesis that individuals with elevated preoperative emotional stress present higher postoperative pain levels was confirmed. Third, the persistent postoperative movement-evoked pain in 48 hours is associated with poor postoperative rehabilitation.

**Table 4. Coefficients from generalized linear model of preoperative emotional stress (B-MEPS) and preoperative predictors.**

| | B | SE | Wald Chi-square | *p* value |
|---|---|---|---|---|
| Schooling | 0.01 | 0.01 | 0.70 | 0.40 |
| Psychiatric diagnosis[#] | 0.37 | 0.14 | 6.55 | 0.012 |
| Cancer diagnosis | -0.02 | 0.12 | 0.03 | 0.84 |
| Preoperative pain | 0.10 | 0.14 | 0.51 | 0.47 |
| CSI | 0.02 | 0.003 | 46.74 | < 0.01 |
| Gender[&] | 0.01 | 0.19 | 0.00 | 0.99 |
| BDNF | 0.07 | 0.0 | 4.42 | 0.78 |
| Gender*BDNF | 0.007 | 0.004 | 0.25 | 0.61 |
| S100B | 0.0 | 0.1 | 0.44 | 0.50 |
| CPM | -0.04 | 0.02 | 0.03 | 0.86 |

[&]Estimate means for B-MEPS: male = -0.11, female = -0.14 (Mean difference = 0.025; *p* = 0.83).

[#]Estimate means for B-MEPS according to psychiatric diagnosis: presence = 0.05 (0.13); absence = -0.32 (0.08); *p* = 0.01. SE, Standard Error; CSI, Central Sensitivity Inventory; BDNF, brain-derived neurotrophic factor; CPM, Conditioned Pain Modulation.

**Table 5. MANCOVA model for pre- and postoperative pain variables comparing low or high psychological stress according to B-MEPS tool*.**

| Dependent variable | Low stress (SE) | High stress (SE) | F value | p value |
|---|---|---|---|---|
| **Psychophysical pain tests** | | | | |
| Pressure pain threshold | 2.79 (0.29) | 3.11 (0.31) | 0.89 | 0.34 |
| Pressure pain tolerance | 7.46 (0.28) | 7.59 (0.68) | 0.03 | 0.86 |
| CPM | -1.38 (0.24) | -1.91 (0.59) | 0.66 | 0.41 |
| **Postoperative pain** | | | | |
| Movement-evoked pain 12 h | 5.44 (0.27) | 6.88 (0.68) | 4.02 | 0.047 |
| Movement-evoked pain 24 h | 6.12 (0.24) | 7.57 (0.61) | 4.75 | 0.031 |
| Movement-evoked pain 48 h | 5.59 (0.25) | 6.24 (0.63) | 1.01 | 0.31 |
| Pain at rest 12 h | 2.68 (0.26) | 3.94 (0.65) | 3.11 | 0.08 |
| Pain at rest 24 h | 3.38 (0.28) | 4.33 (0.70) | 1.56 | 0.21 |
| Pain at rest 48 h | 2.68 (0.24) | 3.03 (0.61) | 0.27 | 0.60 |

*Adjusted controlling for age, years of study and cancer surgery. B-MEPS, Brief Measure of Emotional Preoperative Stress; SE, Standard Error; CPM, Conditioned Pain Modulation.

Initially in this cohort, we explored predictors of psychological stress and we demonstrated that previous psychiatric diseases and central sensitization (evaluated by CSI) were significant predictors of emotional preoperative stress reflected in the B-MEPS results. Central sensitization is responsible for alterations in pain sensitivity thresholds in acute and chronic pain situations [17]. Therefore, we can assume that sensitized patients have a higher psychological vulnerability, which is associated to higher preoperative emotional stress.

**Table 6. Parameter estimates from repeated measures of Visual analogue scale for movement-evoked pain intensity using mixed model analysis.**

| | Predictors | Movement-evoked pain mean (SD) | F value | Estimate | SE | p value |
|---|---|---|---|---|---|---|
| | Low/intermediate stress | 5.81 (0.17) | 7.26 | | | |
| Model 1 | High stress | 7.14 (0.45) | | | | |
| | B-MEPS | | 7.26 | -1.47 | 0.638 | 0.022 |
| | Time | | 2.86 | -0.95 | 0.876 | 0.278 |
| Model 2 | B-MEPS*Time | | 0.10 | 0.30 | 0.946 | 0.748 |
| | Age | | 0.004 | 0.001 | 0.017 | 0.952 |
| | Schooling | | 0.456 | -0.032 | 0.047 | 0.500 |
| Model 3 | Chronic pain diagnosis | | 4.52 | -0.92 | 0.435 | 0.034 |
| | Psychiatric diagnosis | | 0.91 | -0.40 | 0.424 | 0.340 |
| | Cancer diagnosis | | 6.17 | -0.91 | 0.368 | 0.014 |
| Model 4 | Pressure pain tolerance | | 8.35 | -0.14 | 0.0517 | 0.004 |
| | BDNF | | 3.83 | -0.00 | 0.0026 | 0.051 |
| Model 5 | Neuraxial morphine | | 0.263 | -0.29 | 0.579 | 0.609 |
| | Combined neuraxial versus general anesthesia | | 0.109 | 0.25 | 0.762 | 0.741 |
| | Surgery duration | | 3.27 | 0.35 | 0.193 | 0.071 |

The effect of demographic (age, schooling), preoperative (cancer, previous pain, psychiatric disease), experimental pain tests (pressure pain tolerance), biomarker (BDNF), anesthesia/surgery (type, morphine in neuroaxis, surgical duration) were tested in 5 models: *Model 1*: B-MEPS Category and time; *Model 2*: Model 1 plus age and schooling; *Model 3*: Model 2 plus cancer, chronic pain and psychiatric diagnosis; *Model 4*: Model 3 plus pressure pain tolerance and BDNF; *Model 5*: Model 4 plus neuraxial morphine, anesthesia type, surgery duration. Effect size B-MEPs (standard mean difference) on movement-evoked pain: movement pain high stress-movement pain low stress/ SD = 7.14–5.81/2.83 = 0.46 –moderate effect size. SD, Standard Deviation; SE, Standard Error; B-MEPS, Brief Measure of Emotional Preoperative Stress; BDNF, brain-derived neurotrophic factor.

**Table 7. Generalized linear model with dependent variable morphine consumption in 48 hours.**

|  | Mean (SE) | B (SE) | Wald chi-square | *p* value |
|---|---|---|---|---|
| **Low stress** | 8.10 (0.9) | -3.09 (1.46) | 4.46 | 0.035 |
| **High stress** | 11.46 (1.4) |  |  |  |
| **Gender (male x female)** | 10.86 (1.2) | 1.73 (1.11) | 2.41 | 0.12 |
|  | 9.13 (1.12) |  |  |  |
| **No cancer surgery** | 8.5 (1.09) | 2.92 (1.07) | 7.41 | 0.026 |
| **Cancer surgery** | 11.46 (1.2) |  |  |  |
| **Chronic pain** | 9.75 (1.03) | 0.50 (1.31) | 0.14 | 0.70 |
| **No chronic pain** | 10.25 (1.37) |  |  |  |
| **General anesthesia** | 13.12 (1.87) | 6.23 (2.27) | 7.72 | 0.006 |
| **Regional or combined anesthesia** | 6.88 (1.08) |  |  |  |
| **Neuraxial morphine present** | 9.78 (1.28) | 0.42 (1.75) | 0.05 | 0.80 |
| **No neuraxial morphine** | 10.21 (1.4) |  |  |  |
| **CPM** |  | 0.16 (0.17) | 0.84 | 0.35 |
| **Pressure pain tolerance** |  | 0.01 (0.16) | 0.00 | 0.92 |

SE, Standard Error; CPM, Conditioned Pain Modulation.

However, neither S100B, a central biomarker of neuronal damage, nor BDNF was related to high emotional preoperative stress levels, in spite of the known functional consequences of stress-induced structural plasticity in some brain regions [20].

One explanation could be associated to the acute and transient nature of emotional preoperative stress, in comparison to chronic pain situations where these neuromodulators had a pattern of elevation [21].

The existence of psychological vulnerability in perioperative practice is a concept already established and it can be largely witnessed and observed by experienced practitioners [22]. There are some few instruments designed to evaluate patient psychological profile, such as the hospital anxiety and depression scale (HADS) [23], the Spielberger State Anxiety Inventory (SAI) [24], and Beck Anxiety Inventory (BAI) [25]. Nonetheless, all of them are based on the classical test theory. The B-MEPS tool has some important advantages as a valid screening tool to detect psychological vulnerability in the preoperative setting. It is an instrument developed with powerful response item theory analysis, based on items from four other psychological scales. To begin with, it considers the individual response to each question and its relation to the latent trait stress [16] has been more accurate than a classic test theory based scale [26]. We developed a digital tool that shares an interface with the R statistical program, overcoming

**Table 8. Result for univariate analysis of variance for QoR-15 rehabilitation.**

|  | F | *p* value |
|---|---|---|
| **B-MEPS result (low x high)** | 0.52 | 0.47 |
| **Schooling** | 0.33 | 0.56 |
| **Cancer surgery** | 1.46 | 0.28 |
| **General anesthesia (general x regional)** | 0.001 | 0.96 |
| **Postoperative movement-evoked pain in 24 h** | 1.16 | 0.34 |
| **Postoperative movement-evoked pain in 48 h** | 15.69 | < 0.001 |
| **Surgical duration** | 0.08 | 0.92 |

B-MEPS, Brief Measure of Emotional Preoperative Stress; QoR-15, Quality of Rehabilitation Questionnaire– 15.

what would otherwise be a considerable challenge: performing a calculation based on an item response theory at the patient's bedside (A. C. Schiavo, M.D., unpublished data, December, 2018). Therefore, the B-MEPS provides an effective and rapid screening of the emotional profile related to stress in the perioperative period and it can be used for clinical and research purposes.

Next, our hypothesis that individuals with low capacity to respond to acute and prolonged stressors in a biopsychosocial perspective had a propensity for increased postoperative pain was confirmed. Movement-evoked pain along the first 24 hours was the main pain outcome evaluated in our cohort. Based on a recent systematic review [27] using a percentage ratio, it was found that postoperative movement-evoked pain is 95–226% more intense than pain at rest in the first 3 postoperative days. We found that mean movement-evoked pain in the first 12 to 48 hours was 95–105% higher than pain at rest. A mixed model for repeated measures has shown a sustainable significant effect of B-MEPS as movement-evoked pain predictor independently of demographic data, comorbid conditions, preoperative pain test, type of anesthesia, and surgical duration. We also confirmed previous chronic pain, cancer surgery, pressure pain tolerance and BDNF (with almost statistical significance $p = 0.051$) as independent predictors of postoperative movement-evoked pain. This result illustrates the combination of elements which represent the psychological (B-MEPS), physical (cancer, chronic pain), psychophysical (negative association with pressure pain tolerance) and neurobiological (BDNF) complexity of postoperative pain. High stress also predicted morphine consumption, besides general anesthesia and cancer surgery (Fig 3).

This result confirms that a brief screening method of preoperative emotional state could detect individuals at a high risk of experiencing severe postoperative pain. Moreover, morphine consumption in 48 hours was the only significant predictor of poor rehabilitation measured by QoR-15 questionnaire. Our study showed an unexpected high frequency of severe postoperative movement-evoked pain (Numeric Pain Scale (NPS) $\geq 7$) reaching 53% in 24 hours and 42% in 48 hours (Fig 4), considering that the majority of patients received combined or regional anesthesia in a hospital with Acute Pain Service Team 24/7. This finding confirms the fragility of pain at rest evaluation as the fifth vital sign and the gap between medical prescription and patient expectation or perception, despite greater awareness and clinical advancements in pain management.

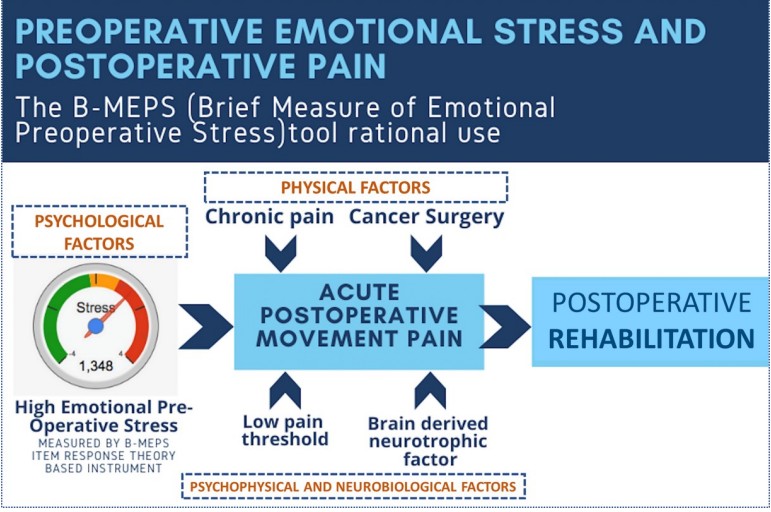

**Fig 3. Infographic of preoperative emotional stress and postoperative pain.**

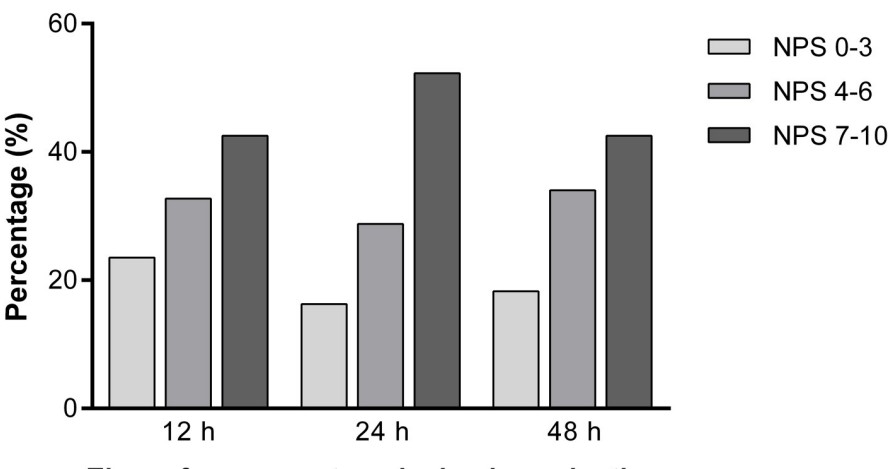

**Fig 4. Acute movement-evoked postoperative pain at 12, 24 and 48 hours.** NPS 0–3: low pain, NPS 4–6: moderate pain, NPS 7–10: severe pain. NPS, Numeric Pain Scale.

Psychophysiological predictors for postoperative acute or chronic pain have been the focus of extensive perioperative research. In a qualitative systematic review including 48 studies, preoperative pain, anxiety, age, and type of surgery were four significant predictors for postoperative pain [28]. A cohort of 1490 patients undergoing heterogeneous surgical procedures recorded their pain until 5 days after surgery, pointing out that the most important pain predictors are preoperative pain, expected pain, surgical fear, and pain catastrophizing [8]. A systematic review of experimental pain studies revealed that quantitative sensory testing may predict up to 54% of the variance in postoperative pain experience [29]. Grosen reported that CPM predicted morphine consumption and situational pain catastrophizing predicted movement-evoked pain intensity in patients submitted to chest wall surgery [30]. In our cohort we did not find association between CPM and stress or postoperative pain, which could be explained by the small percentage of patients with chronic pain or central sensitization.

Our study has some limitations. Firstly, the postoperative pain evaluation by the unidimensional numeric pain scale does not entirely reflect the multidimensional aspects of postoperative pain. Low pain scores do not guarantee that patients feel comfortable or able to perform the necessary activities to expedite recovery outcomes, neither high scores are always associated to poor patient's opinion on the acceptability of pain [31]. Secondly, our follow-up is limited to 48 hours and we did not evaluate postoperative complications. Nevertheless, our focus is to deeply understand the psychological vulnerability influence on postoperative pain, considering that adequate pain management is intrinsically linked to rehabilitation [32] and the reduction of complications [33]. Lastly, in a clinical study, it is not possible to directly assess and isolate the effect of each potential confounding factor of the dependent variables, especially because the number of predictors is high for such complex pain-related outcomes.

The next step to translate this finding into possible beneficial changes in the perioperative assistance is to plan specific interventions considering the emotional preoperative stress. Several strategies could be included focusing on patient-centered outcomes. For example, simple organizational approaches based on caregiver empathic behavior reduced anxiety at the preoperative consultation visit and, thus, increased patient satisfaction after surgery [34]. In a prospective three-arm randomized clinical trial with a 6-month follow-up, 124 patients scheduled for cardiac artery bypass graft (CABG) surgery were randomized to either a brief psychological pre-surgery intervention to optimize outcome expectations (EXPECT) or a psychological

control intervention focusing on emotional support rather than on expectations (SUPPORT); or to a standard medical care (SMC). The intervention group was encouraged to develop personal ideas and images about their future focusing on the development of realistic expectations about the benefits of the surgery and the recovery process. They had a better mental quality of life and fitness for work within six-months and less pro-inflammatory cytokines in the postoperative period [35]. More specifically, the Toronto General Hospital, in order to implement better outcomes related to pain, incorporated a multidisciplinary program focused on the early identification of patients at risk for chronic pain after surgery. They offered coordinated and comprehensive care consisting of pain physicians, nurses, psychologists, and physiotherapists, which grants the opportunity to impact patients' pain trajectories, preventing the transition from acute to chronic pain, and reducing suffering, disability, and health care costs [36].

Elevated emotional preoperative stress levels assessed by B-MEPS results could also be a trigger to individualized pharmacological and non-pharmacological interventions in order to regulate the homeostasis between different systems such as endocrine, nervous, and immune systems [14].

## Conclusions

Our results confirm that B-MEPS is a consistent method for screening preoperative emotional status that can detect individuals prone to moderate to severe postoperative pain. To translate these findings into beneficial changes in the perioperative care, non-pharmacological interventions such as emotional preparation, improvement in communication and patient support, and even pharmacological interventions according to the level of emotional stress should be tested.

## Supporting information

**S1 Table. Complete database of B-MEPS study.**
(XLSX)

## Acknowledgments

Research reported in this publication was supported by Research and Events Incentive Fund—Hospital de Clínicas de Porto Alegre (FIPE-HCPA), Porto Alegre, Brazil; Postgraduate Program in Medical Sciences, School of Medicine, Universidade Federal do Rio Grande do Sul, Porto Alegre, Brazil; Postgraduate Research Group, HCPA, Porto Alegre, Brazil. The funding sources had no influence in the study design, data collection and analyses, manuscript preparation, or in the decision to submit the article for publication.

## Author Contributions

**Conceptualization:** Anelise Schifino Wolmeister, Carolina Lourenzon Schiavo, Andressa de Souza, Wolnei Caumo, Luciana Cadore Stefani.

**Data curation:** Stela Maris de Jezus Castro.

**Formal analysis:** Anelise Schifino Wolmeister, Stela Maris de Jezus Castro, Luciana Cadore Stefani.

**Funding acquisition:** Luciana Cadore Stefani.

**Investigation:** Anelise Schifino Wolmeister, Carolina Lourenzon Schiavo, Kahio César Kuntz Nazário, Rafael Poli Caetani.

**Methodology:** Anelise Schifino Wolmeister, Carolina Lourenzon Schiavo, Kahio César Kuntz Nazário, Andressa de Souza, Rafael Poli Caetani, Wolnei Caumo, Luciana Cadore Stefani.

**Project administration:** Luciana Cadore Stefani.

**Resources:** Anelise Schifino Wolmeister, Carolina Lourenzon Schiavo, Kahio César Kuntz Nazário, Rafael Poli Caetani.

**Software:** Stela Maris de Jezus Castro.

**Supervision:** Wolnei Caumo, Luciana Cadore Stefani.

**Writing – original draft:** Anelise Schifino Wolmeister.

**Writing – review & editing:** Anelise Schifino Wolmeister, Luciana Cadore Stefani.

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
