## [Decision Letter · Decision Letter 0]

22 Oct 2019

PONE-D-19-24073

The Brief Measure of Emotional Preoperative Stress (B-MEPS) as a new predictive tool for postoperative pain: A prospective observational cohort study

PLOS ONE

Dear Professor Stefani,

Thank you for submitting your manuscript to PLOS ONE. After careful consideration, we feel that it has merit but does not fully meet PLOS ONE’s publication criteria as it currently stands. Therefore, we invite you to submit a revised version of the manuscript that addresses the points raised during the review process

We would appreciate receiving your revised manuscript by Dec 06 2019 11:59PM. Wen you are ready to submit your revision, log on to https://www.editorialmanager.com/pone/ and select the 'Submissions Needing Revision' folder to locate your manuscript file.

To enhance the reproducibility of your results, we recommend that if applicable you deposit your laboratory protocols in protocols.io, where a protocol can be assigned its own identifier (DOI) such that it can be cited independently in the future. For instructions see: http://journals.plos.org/plosone/s/submission-guidelines#loc-laboratory-protocols

We look forward to receiving your revised manuscript.

Kind regards,

Yan Li

Academic Editor

PLOS ONE

**Journal Requirements:**

**Additional Editor Comments (if provided):**

Thank you for submitting your manuscript "The Brief Measure of Emotional Preoperative Stress (B-MEPS) as a new predictive tool for postoperative pain: A prospective observational cohort study" to PlosOne,

we would like to invite the authors to revise the manuscript before we finally accept it.

**Comments to the Author**

1. Is the manuscript technically sound, and do the data support the conclusions?

Reviewer #1: Yes

Reviewer #2: Partly

2. Has the statistical analysis been performed appropriately and rigorously? 

Reviewer #1: Yes

Reviewer #2: I Don't Know

3. Have the authors made all data underlying the findings in their manuscript fully available?

Reviewer #1: Yes

Reviewer #2: Yes

4. Is the manuscript presented in an intelligible fashion and written in standard English?

Reviewer #1: Yes

Reviewer #2: Yes

5. Review Comments to the Author

Reviewer #1: In this manuscript, the authors developed an evaluation system to assess the stress level of postoperative patients. However, a though review of the existing methods, and the pros and cons seems essential. Given more evidence that the existing ones might not meet the needs, and the new one is beneficial can make this method more convincing and more acceptable.

Reviewer #2: The manuscript presents data that supports the conclusions. The title, abstract, introduction and discussion all focus on "whether preoperative emotional stress evaluated with a new tool - the B-MEPS - can predict postoperative outcomes". This question is addressed by each section of the manuscript.

However, there is additional data that is not mentioned in the abstract and is not introduced until the last paragraph of the introduction. The aim "to investigate the possible neurobiological or neurophysiological mechanisms implicated in high preoperative stress" is not as fully addressed by the manuscript. Options would include:

a. include more information relating to this aim in the abstract, introduction and discussion, or

b. remove this data to publish separately.

My opinion is that this aspect of the study would be better presented separately. This would also shorten the results section making the manuscript easier to read.

I have said "yes" to the question "Is the manuscript presented in an intelligible fashion and written in standard English?". The manuscript is written in English that is a little stilted in places but I think the meaning is apparent.

Positive aspects of this manuscript include:

- the primary question is of relevance for clinicians and patients (and I found it interesting).

- the B-MEPS appears to have been pragmatically designed and as such is a potential clinical tool rather than simply a research interest.

- the title and abstract are clear and easily understandable.

- the discussion and conclusions are reasonable.

- the discussion commented on several points that add to the understanding of perioperative pain research in general e.g. "the fragility of rest pain evaluation as the fifth vital sign".

- potential pre-optimisation with psychological preparation is a "hot topic" in preoperative medicine currently and this ties in nicely.

I found table 3 helpful. To me, tables 5-7 added less to the clarity of the manuscript than tables 1-4.

I hope this is of some help. If required I can expand on these comments.

6. PLOS authors have the option to publish the peer review history of their article (what does this mean?). If published, this will include your full peer review and any attached files.

Reviewer #1: No

Reviewer #2: Yes: Kate Chatten

---

## [Author Response · Author response to Decision Letter 0]

12 Dec 2019

Thanks in advance for the reviewers’ comments. The answers are highlighted below each comment.

Review Comments to the Author

Reviewer #1:

In this manuscript, the authors developed an evaluation system to assess the stress level of postoperative patients. However, a though review of the existing methods, and the pros and cons seems essential. Given more evidence that the existing ones might not meet the needs, and the new one is beneficial can make this method more convincing and more acceptable.

Response: Thank you for your comments. We made some adjustments in the introduction to highlight the possible benefits of our new instruments, a broad emotional evaluation of the preoperative period. The sentences below were included in the introduction.

“Nevertheless, analyzing, in a practical and consistent way, a patient's psychological profile in the preoperative period constitutes a challenging task, even considering the existing evidence of worst rehabilitation outcomes related to catastrophizing levels [6], anxiety [7], surgical fear [8] and preoperative pain [8,9]. In general, a good psychological state is a health indicator. Healthy psychological aspects such as life satisfaction, optimism, self-esteem, and perception of social support can positively influence several health indices. On the other hand, factors such as depression, anxiety, hostility reflect a less desirable psychological state, which may influence short and long-term recovery in direct and indirect ways. The direct effect is related to the impact of emotions on the stress hormones (cortisol, adrenaline, noradrenaline) which regulate healing and many physiological responses. Besides, the indirect effect of psychological burden can be reflected on behavioral responses to stress such as poor self-care, smoking, alcohol intake, anxiety, depression and sleep deprivation [10–12].”

Therefore, more than being a sole psychological scale designed to measure just one symptom (depression or anxiety), the B-MEPS instrument aims at adding a broad emotional evaluation to the preoperative setting. Moreover, the purpose of measuring an emotional construct is to evaluate its impact on perioperative outcomes.

Reviewer #2: 

The manuscript presents data that supports the conclusions. The title, abstract, introduction and discussion all focus on "whether preoperative emotional stress evaluated with a new tool - the B-MEPS - can predict postoperative outcomes". This question is addressed by each section of the manuscript.

However, there is additional data that is not mentioned in the abstract and is not introduced until the last paragraph of the introduction. The aim "to investigate the possible neurobiological or neurophysiological mechanisms implicated in high preoperative stress" is not as fully addressed by the manuscript. Options would include: 

a. include more information relating to this aim in the abstract, introduction and discussion, or

b. remove this data to publish separately. 

My opinion is that this aspect of the study would be better presented separately. This would also shorten the results section making the manuscript easier to read.

I have said "yes" to the question "Is the manuscript presented in an intelligible fashion and written in standard English?". The manuscript is written in English that is a little stilted in places but I think the meaning is apparent.

Positive aspects of this manuscript include:

- the primary question is of relevance for clinicians and patients (and I found it interesting).

- the B-MEPS appears to have been pragmatically designed and as such is a potential clinical tool rather than simply a research interest.

- the title and abstract are clear and easily understandable.

- the discussion and conclusions are reasonable.

- the discussion commented on several points that add to the understanding of perioperative pain research in general e.g. "the fragility of rest pain evaluation as the fifth vital sign".

- potential pre-optimisation with psychological preparation is a "hot topic" in preoperative medicine currently and this ties in nicely.

I found table 3 helpful. To me, tables 5-7 added less to the clarity of the manuscript than tables 1-4.

I hope this is of some help. If required I can expand on these comments.

Response: Thank you for your careful review. Regarding your suggestions we can offer an explanation. 

As an exploratory study we intended to evaluate all the aspects possibly involved in emotional stress and its consequences in acute postoperative pain and rehabilitation. In view of a relatively new proposal of preoperative construct, the emotional stress, it sounded better to present the rationale of the study stepwise. In other words, from the factors possibly associated with the emotional stress to its consequences regarding acute postoperative pain and rehabilitation.

Our idea was to maintain the exploratory analysis of the possible mechanisms of emotional stress. Hence the inclusion of some sentences in the abstract, introduction and discussion, as you suggested.

Abstract: “Moreover, the possible neurobiological or neurophysiological mechanisms implicated in high preoperative emotional stress, evaluated through preoperative quantitative sensory pain tests d seanrum biomarkers BDNF and S100B were investigated.”

Introduction: “A huge variation in postoperative pain thresholds is frequently observed in the postoperative scenario, even for similar surgical trauma and type of anesthesia. Preoperative patients’ vulnerabilities such as physical, social, and psychological are implicated in this variability [5]. Nevertheless, analyzing, in a practical and consistent way, a patient’s psychological profile in the preoperative period constitutes a challenging task, even considering the existing evidence of worst rehabilitation outcomes related to catastrophizing levels [7], anxiety [8], surgical fear [9] and preoperative pain [9,10]. In general, a good psychological state is a health indicator. Healthy psychological states such as life satisfaction, optimism, self-esteem, and perception of social support can positively influence several health indices. On the other hand, factors such as depression, anxiety, hostility reflect a less desirable psychological state, which may influence short and long-term recovery in direct and indirect ways. The direct effect is related to the impact of emotions on the stress hormones (cortisol, adrenaline, noradrenaline) which regulate healing and many physiological responses. Besides, the indirect effect of psychological burden can be reflected on behavioral responses to stress such as poor self-care, smoking, alcohol intake, anxiety, depression and sleep deprivation.”

Discussion: “However, neither S100B, a central biomarker of neuronal damage, nor BDNF was related to high emotional preoperative stress levels, in spite of the known functional consequences of stress-induced structural plasticity in some brain regions [20].

One explanation could be associated to the acute and transient nature of emotional preoperative stress, in comparison to chronic pain situations where these neuromodulators had a pattern of elevation [21].”

Therefore, more than being a sole psychological scale designed to measure just one symptom (depression or anxiety), the B-MEPS instrument aims at adding a broad emotional evaluation to the preoperative setting. However, the neurophysiological and biological mechanisms possibly related to the emotional preoperative stress and its consequences need to be explored for a better understanding of the preoperative patient’s profile as a whole.

We also added an infographic to the discussion to illustrate the combination of elements which represent the psychological (B-MEPS), physical (cancer, chronic pain), psychophysical (negative association with pressure pain tolerance) and neurobiological (BDNF) complexity of postoperative pain.

---

## [Editor Report · Decision Letter 1]

19 Dec 2019

The Brief Measure of Emotional Preoperative Stress (B-MEPS) as a new predictive tool for postoperative pain: A prospective observational cohort study

PONE-D-19-24073R1

Dear Dr. Stefani,

We are pleased to inform you that your manuscript has been judged scientifically suitable for publication and will be formally accepted for publication once it complies with all outstanding technical requirements.

With kind regards,

Yan Li

Academic Editor

PLOS ONE
---

## [Editor Report · Acceptance letter]

27 Dec 2019

PONE-D-19-24073R1 

The Brief Measure of Emotional Preoperative Stress (B-MEPS) as a new predictive tool for postoperative pain: A prospective observational cohort study 

Dear Dr. Stefani:

I am pleased to inform you that your manuscript has been deemed suitable for publication in PLOS ONE. Congratulations! Your manuscript is now with our production department. 

With kind regards,

on behalf of

Dr. Yan Li 

Academic Editor

PLOS ONE